# Glass-cutting medical images via a mechanical image segmentation method based on crack propagation

Yaqi Huang [1,2,3✉], Ge Hu[1,2,3], Changjin Ji[1,2] & Huahui Xiong[1,2]

Medical image segmentation is crucial in diagnosing and treating diseases, but automatic segmentation of complex images is very challenging. Here we present a method, called the crack propagation method (CPM), based on the principles of fracture mechanics. This unique method converts the image segmentation problem into a mechanical one, extracting the boundary information of the target area by tracing the crack propagation on a thin plate with grooves corresponding to the area edge. The greatest advantage of CPM is in segmenting images involving blurred or even discontinuous boundaries, a task difficult to achieve by existing auto-segmentation methods. The segmentation results for synthesized images and real medical images show that CPM has high accuracy in segmenting complex boundaries. With increasing demand for medical imaging in clinical practice and research, this method will show its unique potential.

---

[1] School of Biomedical Engineering, Capital Medical University, Beijing, China. [2] Beijing Key Laboratory of Fundamental Research on Biomechanics in Clinical Application, Capital Medical University, Beijing, China. [3] These authors contributed equally: Yaqi Huang, Ge Hu. ✉email: yqhuang@ccmu.edu.cn

With the rapid development of modern imaging techniques, medical image segmentation plays an increasingly important role in the diagnosis and treatment of various diseases[1]. When computer models are used to simulate physiological phenomena, explore pathogenesis, and design personalized surgery, image segmentation is an essential step for reconstructing the anatomical structure of relevant tissues and organs[2–4]. Some typical segmentation technologies, such as the active contour model[5–10], atlas-based registration[11–14], and neural network-based segmentation[15–18], have become more mature over the past several decades. Other strategies, such as fuzzy clustering[19], the superpixel method[20,21], and graph-cut method[22,23], are also well applied to medical image segmentation. These auto-segmentation methods have promoted the development of medical imaging-based diagnostic and treatment techniques. However, automatic image segmentation of certain complex images, such as soft tissues with blurred and discontinuous boundaries, remains challenging today. A general solution for the myriad of complex medical images that need to be segmented is still wanting. As a result, many complex medical images still rely on manual segmentation, greatly limiting the applications of medical imaging in a wider range of medical study and clinical practice.

One example of a segmentation problem that is difficult to solve can be found in head and neck magnetic resonance (MR) images. We see that the epimysium appears as a white bright line between dark-gray muscle tissues, and the grayscale color of these muscles is quite similar[24]. This dividing line between the muscles may be fuzzy and even discontinuous. No existing automatic methods can identify such a boundary correctly, and it currently must be segmented manually. So the question is, can we segment it automatically with the limited information that we have?

At present, the field of segmentation method study focuses on incremental improvements of existing mature algorithms according to characteristics of the target region[7–10]. Although these improvements can enhance the performance of segmentation methods to some extent, it is difficult to achieve a real breakthrough while being trapped in the existing frameworks. We propose a fresh way to think about the problem: can we solve the segmentation problem easily by transforming it into a problem in another field, such as a mechanics problem? This may sound farfetched because image processing and mechanics seem completely unrelated. But consider the following well-known real-life phenomenon: after we scored a glass surface using a glass cutter, the glass will crack along the scored line under an appropriate load. Therefore, if we convert the grayscale image into a thin plate and transform the boundary line between tissues into a groove or crack on its surface, then we can make this plate fracture along the groove just like cutting glass. Thus, the image segmentation problem transforms into a mechanical calculation of crack propagation on a thin plate. Compared with boundary-overflow issues caused by other algorithms, such as level sets, this mechanical method has an outstanding advantage when there are discontinuities or small grayscale gradients at the edge of the target area in the image. Due to stress concentration at its tip, when subject to external load, the crack can penetrate small gaps of groove-free regions in the thin-plane structure to connect to the crack on the other side of the gap, forming a continuous trace representing the boundary that we want to segment.

Based on the above analysis, we propose a unique method, named the crack-propagation method (CPM), for image segmentation. The core idea of this method is to transform the image segmentation problem into a mechanical problem of crack propagation on a thin plate. Using the principles of fracture mechanics, we can obtain the boundary coordinates of the target area in an image by tracing the cracks along the edge of the relevant region in a thin-plate model. This paper establishes this unique method, and demonstrates its great advantage and potential in image segmentation by segmenting synthetic and realistic medical images, especially those including soft tissues with blurred and broken boundaries.

## Results

**Conversion of grayscale image to a mechanical model**. The term "image" used throughout the paper denotes a two-dimensional image slice if not specified. The first step to transform an image-segmentation to a mechanical fracture problem is to convert the grayscale image to a mechanical model. In this process, a two-dimensional image is transformed into a thin plate with the same size as the image and a thickness that varies with position according to the grayscale value of the pixel at that given location in the image.

The basic principle for converting grayscale to plate thickness is to transform the boundaries of the target region into grooves on the surface of a plate. Under external load, stress will concentrate at the groove, and the material cracks when the stress value exceeds a certain threshold[25,26]. By extracting coordinates of the crack and mapping it onto the original image, we can obtain the boundary of the target area. Here we construct the geometry of the thin plate based on not the grayscale value itself, but the gradient of the grayscale image, which can transform the boundary into grooves because the edge of the object region generally has substantial variation in grayscale level, and therefore a higher gradient, as shown in Fig. 1a.

The deeper the groove, the higher the stress concentration and the greater the tendency of crack propagation along the groove, and therefore the higher the accuracy of the segmentation result. Thus, when transforming an image into a plate, the height difference between the groove and its surrounding structures should be increased as much as possible. In order to increase the height difference, we can increase the grayscale contrast of the image. Figure 1b shows the results from linear and nonlinear transformations. Although a linear transformation can increase the overall grayscale contrast, the left boundary of the image is still blurred. When we perform a nonlinear transformation, the left boundary is further enhanced. We can achieve a much higher overall contrast compared to using a linear transformation, and additionally the grooves formed in the plate model are deeper.

The models established above are one-sided models, characterized by a flat lower and a curved upper surface with grooves. Because the stress distribution on two sides is different, the crack generated on the lower surface does not necessarily coincide with the grooves on the upper surface, which may cause deviations in the path of crack propagation. Figure 1c shows a symmetric model with grooves on the both sides of the thin plate. In the double-sided model, both surfaces reflect the same grayscale gradient, and therefore we can avoid the problem caused by an asymmetric stress distribution along the thickness of the plate in the one-sided model. Therefore, the results of the double-sided model can be more consistent with the boundary of the target area in the image.

**Stress concentration and crack propagation**. The key for image segmentation using CPM is that the crack propagation in the plate due to stress concentration matches the boundary of the target area in the original image. Figure 2a shows the cross-sectional stress distribution under a distributed pulling force perpendicular to the groove in the plate, showing a strong stress concentration at the bottom of the groove. As shown in Fig. 2b, the deeper the groove, the greater the maximum stress. Therefore, when the boundary of the target area on the image is converted

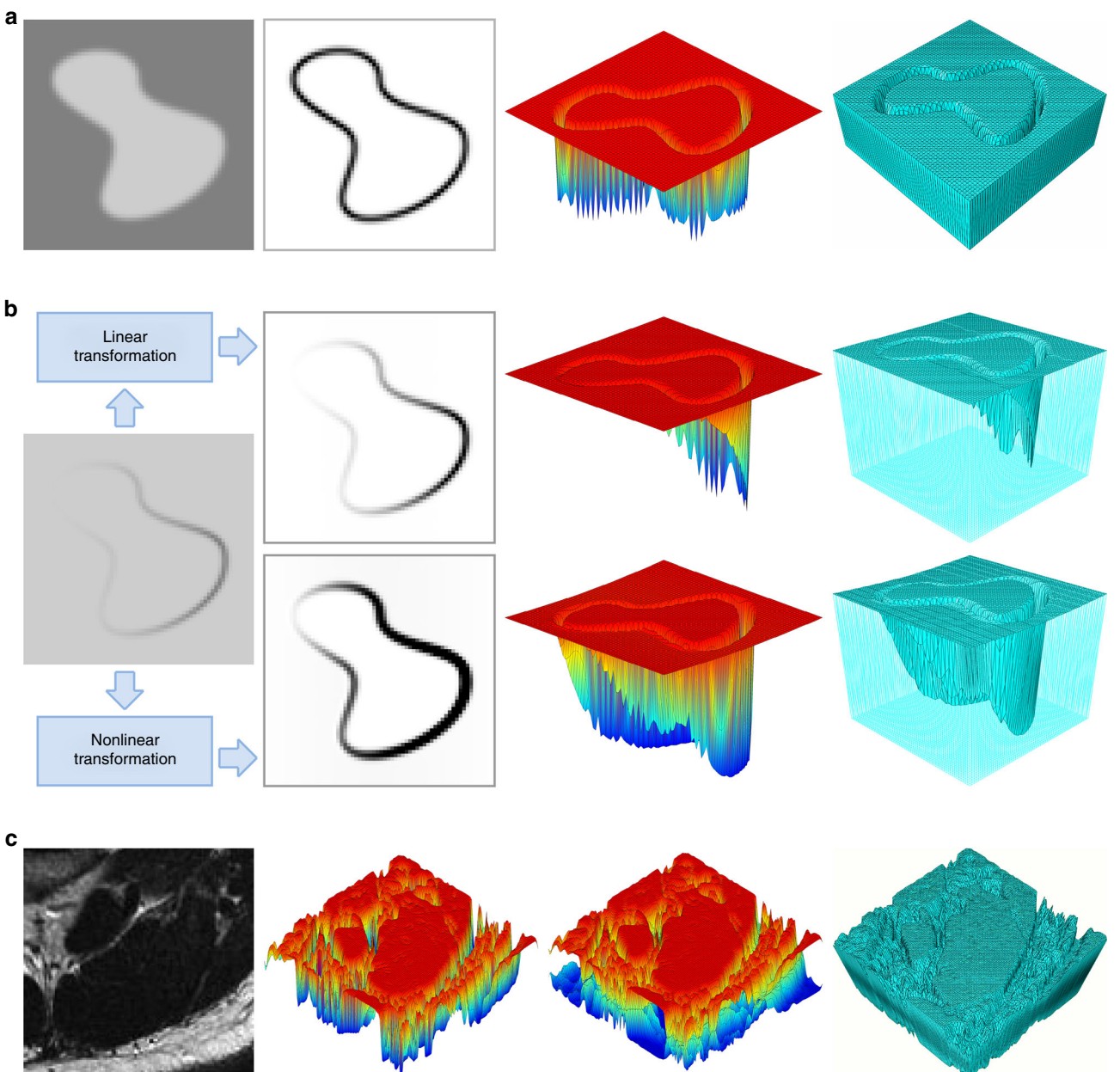

**Fig. 1 Establish mechanical models from images. a** From left to right: original image, gradient image, grayscale trend of the gradient image, and mechanical model established from the gradient image. **b** From left to right: original image, images after intensity transformation, grayscale trend of the images in the second column, and perspective mechanical models corresponding to those same images. **c** From left to right: magnetic resonance image of head and neck muscles, grayscale trend of the one-sided model, grayscale trend of the two-sided model, and bilateral symmetric mechanical model.

into a groove on a plate, the stress at the groove position will be much higher than in other areas under external load.

For the brittle materials, damage and breakage will occur at the location where the stress value reaches the strength limit of the material[27]. The stress concentration at the groove region allows it to reach the damage threshold first, causing it to crack. The crack continues to expand along the groove, tracing a path reflecting the boundary position. Figure 2c is a mechanical model for a thin plate with an arc-like groove. Under the external load **F** near the groove, stress concentration occurs at the groove region marked by the red circle. The local stress value increases rapidly as **F** increases, and a crack is generated there when the stress limit is reached (Fig. 2d). Persistently applying tensile force, the crack will continuously extend along the groove due to stress concentration, and finally will trace along the entire arc of the groove (Fig. 2e).

After mapping the crack back to the original image, we can obtain the coordinates of the arc on the two-dimensional plane (Fig. 2f).

**Segmentation of discontinuous boundaries**. We want to emphasize that crack propagation does not require a fully continuous groove. The greatest advantage of CPM is in dealing with the segmentation of discontinuous boundaries. Discontinuous boundaries are common in complex medical images, and existing segmentation algorithms generally tend to overflow when dealing with weak or discontinuous boundaries, ultimately resulting in incorrect segmentation. Here we show that when the image with a discontinuous boundary is transformed into a thin plate by CPM, the cracks on either side of a discontinuous region can expand and connect through the nongrooved area, finally meeting

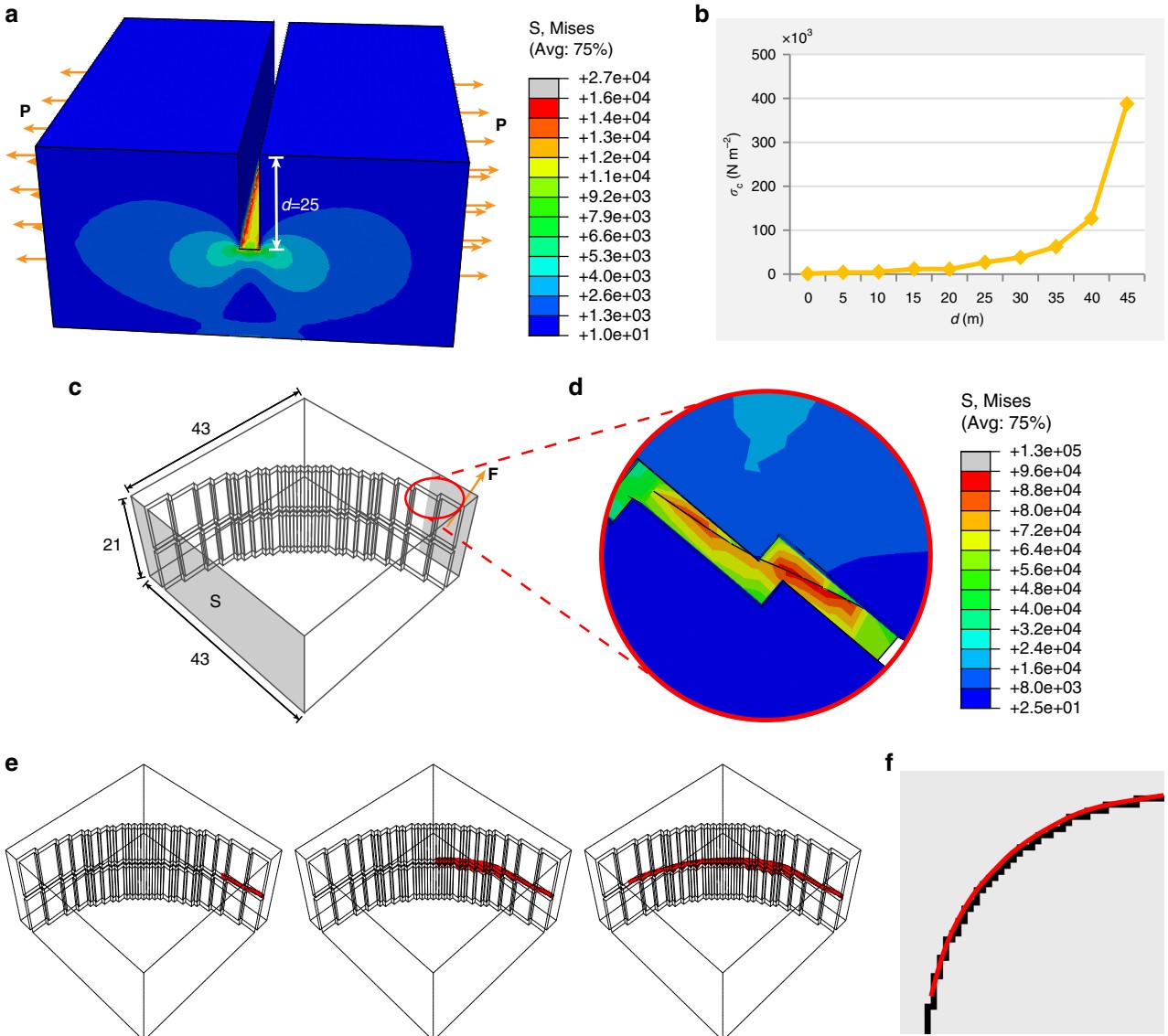

**Fig. 2 Stress concentration and crack propagation. a** The stress distribution of a three-dimensional model under a distributed pulling force **P** perpendicular to the groove. **b** The trend of the maximum stress as the groove-depth increases of the model in **a**. **c** A thin- plate model with a groove transformed from a quarter-circle boundary. The shadow surface S on the left is fixed. **d** The stress distribution and the generated crack of the model enclosed by the red circle in **c** under external load **F**. **e** Crack-surface propagation process in the model. **f** The final boundary obtained after a crack-propagation procedure shown in **e**. Different colors in the scale bar in **a**, **d** represent different stress values (N m$^{-2}$).

together to form a continuous crack corresponding to the correct boundary of the target area.

The strategy to crack a discontinuous region depends on the length of the discontinuity. For a short gap, enclosed by the cyan box in Fig. 3a, the original crack is able to penetrate the gap to continue propagating in the groove on the other side of the gap due to the strong concentrated stress at the crack tip (Fig. 3b). For a wider gap, enclosed by the magenta line in Fig. 3a, because the grooves are too far away from each other, the effect of one groove on the opposite side is too weak to guide the direction of crack propagation. For these cases, we can separate this area into two subregions denoted by boxes 2 and 3 in Fig. 3a, and crack propagation can be performed simultaneously from both sides of the discontinuous region as shown in Fig. 3c, d. After the cracks from the two sides meet together, we can integrate them into a single, smooth crack trace through post-processing procedures (Fig. 3e). In the above simulation, we assumed that the discontinuous region does not contain any edge information. In fact, a discontinuous region can usually provide some boundary

information, which can be used to generate shallow grooves during the image-to-board transformation. These shallow grooves make the crack propagation much easier, and the boundary coordinates extracted from the crack are more accurate.

**Synthetic image segmentation**. The crack-propagation simulation requires a starting position on the groove to generate the first crack locally. Once the initial conditions have been set, the crack will propagate following the groove under the external load applied on the tip region of the crack. The detailed process of image segmentations using CPM can be found in the Methods section.

To test the effectiveness of CPM, we first segment some synthetic images shown in Fig. 4, which contain typical shape features, grayscale distributions, and noise.

In this figure, the distributions of the grayscale values in the background of image (a) and the object of image (b) are inhomogeneous. For these kinds of images, it is easy to incorrectly segment

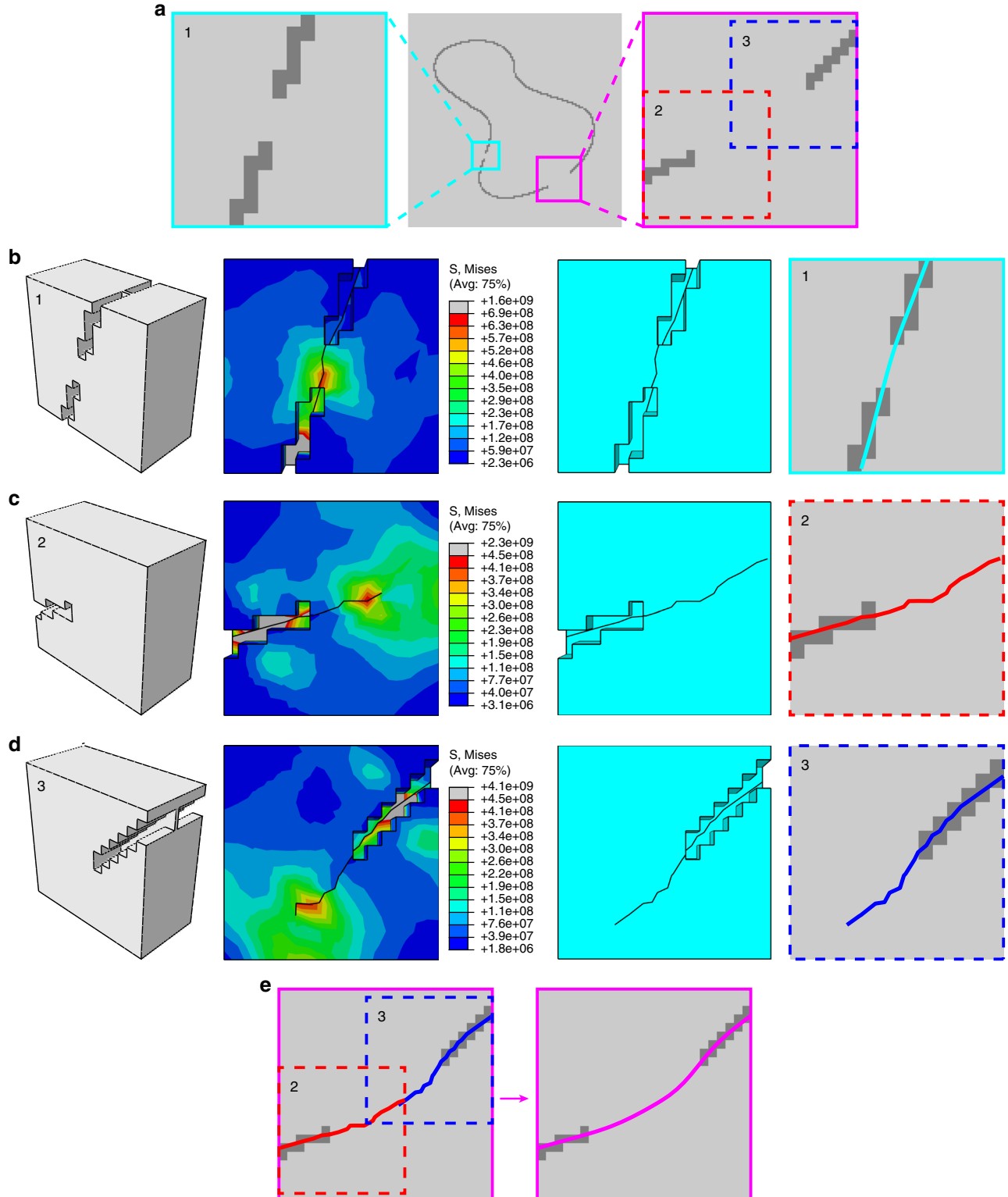

**Fig. 3 Segmentation of discontinuous boundaries. a** An image with discontinuous boundaries. **b** From left to right: mechanical model of the image enclosed by the cyan line in **a**, stress distribution of the model, crack generated in the model, and boundary obtained by the crack. **c** From left to right: mechanical model of the image enclosed by the red dotted line in **a**, stress distribution of the model, crack generated in the model, and boundary obtained by the crack. **d** From left to right: mechanical model of the image enclosed by the blue dotted line in **a**, stress distribution of the model, crack generated in the model, and boundary obtained by the crack. **e** The final boundary of the discontinuous region enclosed by the magenta line after integration of the cracks in **c**, **d**. Different colors in the scale bar in subfigures **b–d** represent different stress values (N m$^{-2}$).

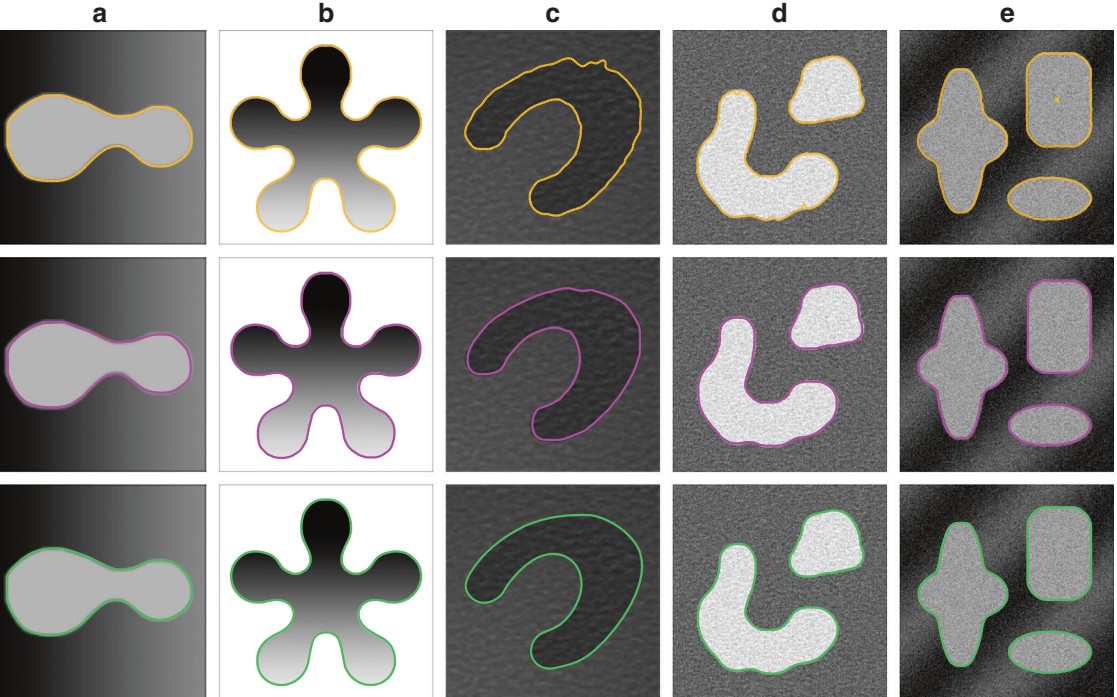

**Fig. 4 Synthetic image segmentation results. a–e** An image with inhomogeneous grayscale distribution in the background, an image with inhomogeneous grayscale distribution in the object, an image with a textured background, a textured image with noise, and an image with both intensity inhomogeneities and Gaussian noise. Yellow curves denote the boundaries obtained by the level-set algorithm, magenta curves represent the final contours obtained by the crack- propagation method, and green curves denote the manual segmentation boundaries.

the boundary at the position where the grayscale value of the object is similar to that of the background using general segmentation methods. However, the small difference in the grayscale value at such locations can be amplified through nonlinear transformations to form grooves in the mechanical model, and therefore the boundary can be identified by the crack propagation. Image (c) is an image with a textured background, and image (d) is a textured image with noise. When we generate mechanical models for these images, the noise may cause unevenness on the surface of the structure. However, the surface fluctuations do not change the overall character of the plate: the stress concentration still manifests at the groove corresponding to the edge, and the generated crack still accurately reflects the right boundary. Image (e) has both intensity inhomogeneities and Gaussian noise. Although it contains complex interference, the correct edge can still be identified using CPM. The first row of Fig. 4 shows the segmentation results using the level-set algorithm[6]. One can see that there is a slight boundary overflow in image (c). The second row is the results using CPM. As a comparison, the third row gives the results from manual segmentation. It is clear that the boundaries segmented using CPM are completely consistent with those obtained by manual segmentation. Therefore, this mechanical method is reliable and accurate for segmenting generic shapes from synthetically generated computer images.

**Medical image segmentation**. The second test is for the segmentations of medical images. To make the test results more representative, we use medical images from different imaging techniques for various human tissues and organs. We collect a total of 67 medical images from MRI, computed tomography (CT), X-ray, and ultrasound, which involve various human tissues and organs, including the bladder, knee, spleen, blood vessels (side view or caliber), kidney, brain tumor, lung, liver, breast tumor, and left ventricle, from published literature and our previous database to perform the test. These images contain different

morphological contours and grayscale features. The results show that both the level-set algorithm and CPM can segment the target boundary well for some images. For other images, CPM can give obviously better segmentation results than the level-set algorithm. Figure 5 shows 11 medical images, taken from each kind of the aforementioned tissues or organs. The first row from image (a) to image (f) in Fig. 5 shows the segmentation results using a typical level-set algorithm DRLSE[6,28], the second row gives the results using CPM, and the third row shows the results from manual segmentations.

The image in Fig. 5a is a bladder MR image[28]. Although most of the outline is clear, the grayscale values of the mid-bottom section are not uniform. Similar to the processing for inhomogeneous images in Fig. 4, we can correctly obtain the boundary information of the fuzzy region by a nonlinear transformation from the image to a thin plate and the simulation of the crack propagation. The second one, image (b), is a knee MR image[29]. Surrounded by other tissues, the target area suffers from interference from the boundaries of other structures. However, although these other structures do generate grooves at their own boundaries, there is little practical effect to the edge detection of the target structure, because the external load is always applied to the positions near the crack tip, and therefore no large stresses will be generated in these neighboring grooves. Image (c) is a CT image of a spleen[30]. There is also interference from the boundaries of other structures, especially at the right boundary of the spleen. However, like in case (b), such interference does not affect the correct boundary detection of the target area. Image (d) is an X-ray image of a blood vessel[28]. This image has a blurred vessel boundary and a complicated contour with low contrast, but it does not affect the generation of the concentrated stress and the propagation of the crack in the grooves transformed from the image. The automatically segmented result using CPM is consistent with the true boundary. Image (e) is an ultrasound image of a blood vessel caliber. Due to the characteristics of

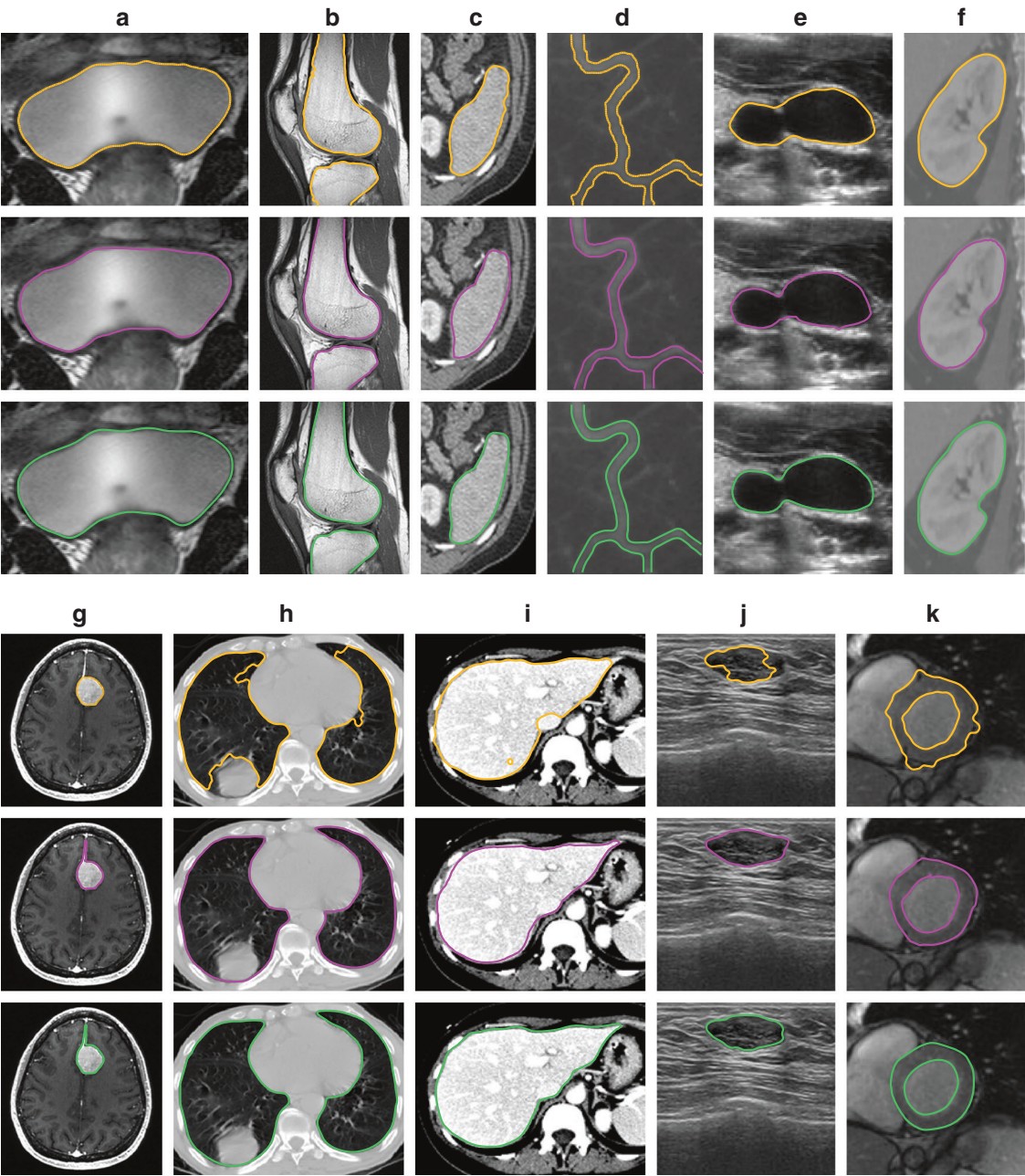

**Fig. 5 Medical image segmentation results. a–k** Magnetic resonance (MR) image of bladder, MR image of knee (reproduced with permission from ref. [29]), computed tomography (CT) image of spleen (reproduced with permission from ref. [30]), X-ray image of blood vessel (reproduced with permission from ref. [28]), ultrasound image of blood vessel caliber, CT image of the kidney (reproduced with permission from ref. [31]), MR image of brain tumor, CT image of lungs (reproduced with permission from ref. [34]), CT image of the liver (reproduced with permission from ref. [35]), ultrasound image of breast tumor (reproduced with permission from ref. [36]), and MR image of the left ventricle. Yellow curves denote the boundaries obtained by the level-set algorithm, magenta curves represent the final contours obtained by the crack-propagation method, and green curves denote the manual segmentation boundaries.

ultrasound imaging, there is fuzziness and noisy background information around the vessel in the image. However, since the grayscale values inside the vessel caliber are low and uniform, significant grayscale changes can form near the boundary. Therefore, both CPM and the level-set algorithm can obtain correct segmentation results. Image (f) is a CT image of a kidney[31]. Similar to image (d), there is a low contrast between the target area and the background in this image. However, the target area is clearly defined and the boundary is not fuzzy, and therefore the kidney boundary can be accurately segmented using both CPM and the level-set algorithm. Image (g) is an MR image

of a brain tumor (https://www.smir.ch/BRATS/Start2016). It is difficult to obtain the exact contour of the elongated structure at the top of the tumor using existing methods, such as the active contour model[32,33], because they often miss boundaries of such fine structure as shown in row 2. However, the CPM proposed in this paper does not have such a limitation. Due to the stress concentration at the groove, the objective edge of the image can be effectively detected by crack propagation. Image (h) is a CT image of the lungs[34]. There is a white area at the bottom of the left lung that is distinctly different from other tissues, and it is difficult to correctly segment this area when using an algorithm

based on region consistency. The segmentation in other parts of the image is also largely affected by interference information in tissues. Because the mechanical method developed in this paper mainly focuses on the structure converted by the grayscale values near the edge, the propagation of the crack is affected by the local stress distribution in the groove, but not the stress in other regions. Therefore, CPM can correctly extract the edge of the interested region. Image (i) is a CT image of a liver[35]. The middle-right section of this image has an interference region similar to image (h), which affects the segmentation accuracy of the level-set algorithm. Image (j) is an ultrasound image of a breast tumor[36]. Different from the ultrasound image (e) for the cross section of a blood vessel, which has uniform grayscale values inside the target area, this image contains substantial transverse texture information within the target area in addition to the interference around the tumor. Therefore, the expansion of the level-set function cannot reach the boundary accurately. Because CPM is not affected by the grayscale distribution far from the target boundary, the segmentation results of CPM are highly consistent with the manually extracted boundary. Figure 5k is an MR image of the left ventricle (http://sourceforge.net/projects/cardiac-mr/files/). Although both the level-set algorithm and CPM can segment the left ventricle, there is some boundary overflow in the level-set algorithm due to the effect of the partial fuzzy boundary. Overall, both CPM and the level-set algorithm can segment images (a)–(f) accurately, and CPM has higher accuracy than the level-set algorithm when segmenting images (g)–(k).

In order to quantitatively compare the performance of CPM and the level-set method, we compare Dice similarity coefficient (DSC)[37] and Hausdorff distance (HD)[38] of the segmentation results for each of the 11 tissue or organ groups in Fig. 6, which involve a total of 67 images (see Source Data). DSC and HD are used to evaluate the similarity and difference between the automatic segmentation results and the manual segmentation boundaries. The average values of the two parameters are 0.9565 and 3.6371 for CPM, and 0.9511 and 7.5555 for the level-set algorithm DRLSE, respectively. The segmentation results of CPM and the level-set algorithm for average DSC are similar, reaching more than 0.95. This means that for most of these medical images, both segmentation algorithms can correctly identify the target area. However, affected by interference information within the target region, the level-set algorithm cannot extend its detection to the correct boundary position in some images, such as those shown in Fig. 5g, k. Therefore, the HD value of segmentation results obtained using the level-set algorithm is higher than that using CPM. The accurate segmentations of these medical images with complex contours and fine structures indicate that CPM may have important applications and bright prospects in medical image-segmentation applications.

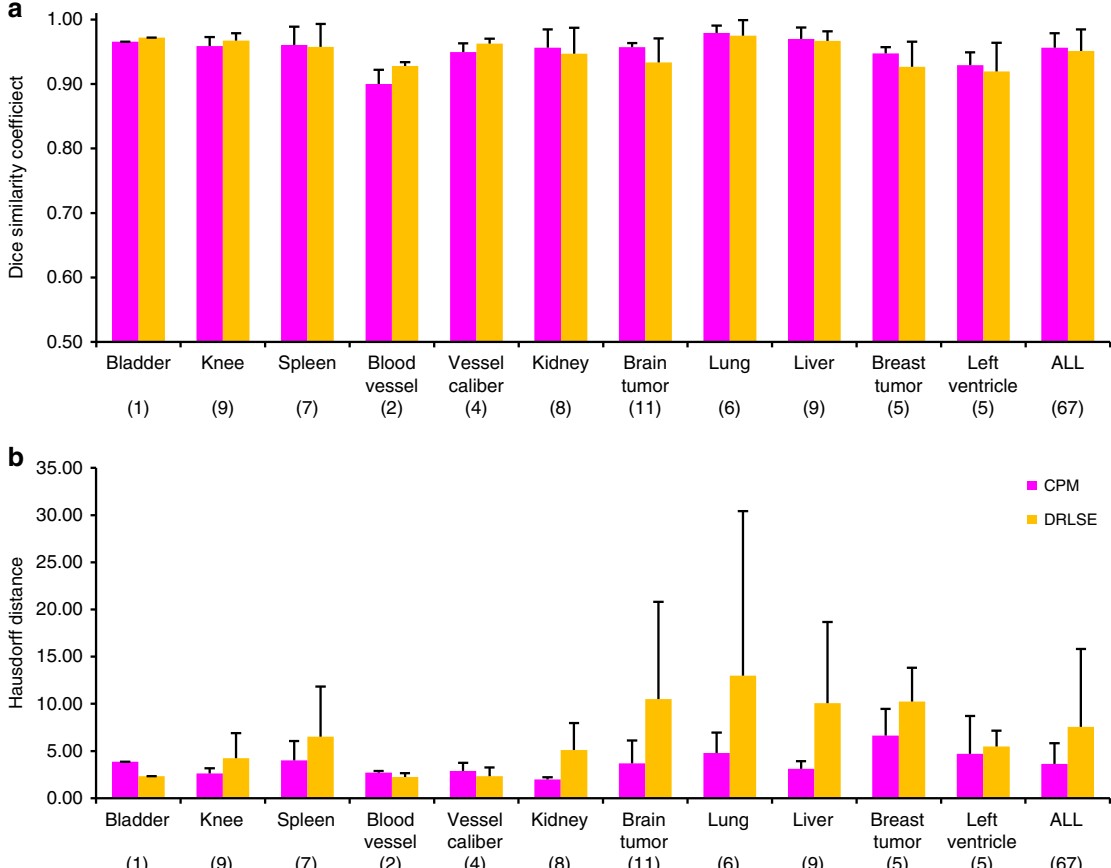

**Fig. 6 Comparison of the crack-propagation method (CPM) and the level-set algorithm DRLSE.** Statistical results for **a** Dice similarity coefficient and **b** Hausdorff distance. The first 11 data groups in **a**, **b** are for each image group, from magnetic resonance (MR) image of the bladder, MR image of the knee, computed tomography (CT) image of the spleen, X-ray image of the blood vessel, ultrasound image of blood vessel caliber, CT image of the kidney, MR image of brain tumor, CT image of lungs, CT image of the liver, and ultrasound image of breast tumor, to MR image of the left ventricle. The last data group combines all 67 images in these 11 groups. The number within the parentheses indicates the number of the images in that group. Magenta bars represent the CPM segmentation results and yellow bars represent the DRLSE. The error bars represent the standard deviation. Source data are provided in the Source Data file.

**Soft-tissue image segmentation.** For images of soft tissues, such as muscle groups, due to similar grayscale values in the different parts of the image, boundaries between different tissues are difficult to distinguish using existing automatic segmentation algorithms. Here we select 60 images of 12 head and neck muscles, including both left- and right sternocleidomastoid (SCM), trapezius (TZ), splenius capitis (SC), semispinalis capitis (SSC), levator scapulae (LS), and obliquus capitus inferior (OCI) with different positions, shapes and sizes, to segment using CPM, and compare the results with that obtained using the level-set algorithm DRLSE.

Because there are many fuzzy or discontinuous boundaries in the images of the head and neck muscles, the level-set algorithm is prone to serious overflow when discovering these boundaries, resulting in incorrect segmentation results. However, although boundaries between the muscles are sometimes blurred, there are still light-colored edges visible to the naked eye. When these visible edges are transformed into grooves to build a mechanical model, the boundaries of the tissues can be identified by crack propagation along the grooves. Furthermore, the technique is able to connect the discontinuous regions of the grooves and arrive at a continuous and complete boundary.

Figure 7 shows partial segmentation results of these soft tissues. The segmentation results of the DRLSE algorithm for the upper-right corner of the obliquus capitus inferior image in (f) and the upper-left corner of the levator scapulae image in (k) contain leakages, and all other tissue images contain significant boundary overflow. Since both the bottom of the obliquus capitus inferior in (l) and the middle of the sternocleidomastoid in (g) have an interference region similar to the CT image of the lungs in Fig. 5h,

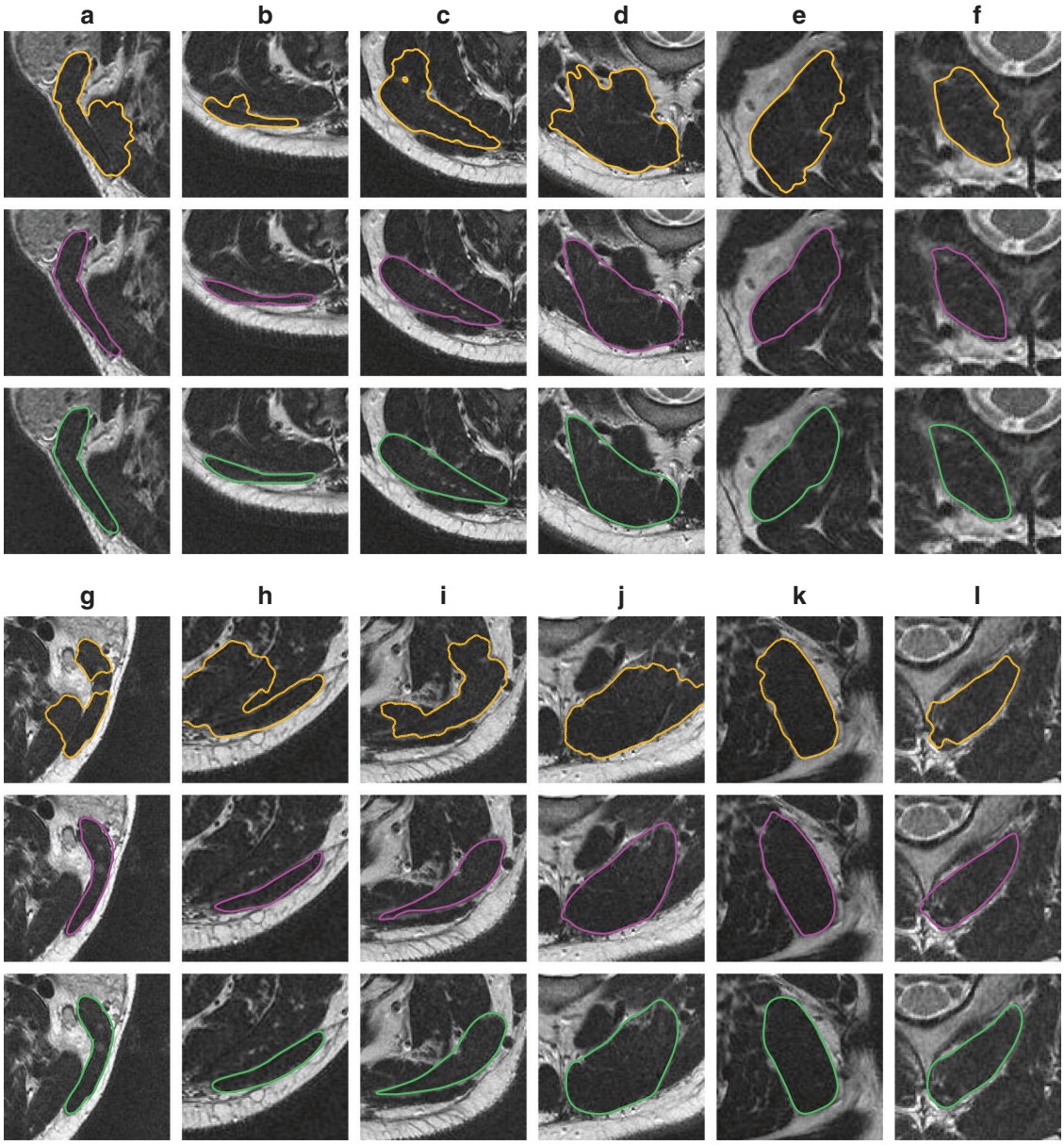

**Fig. 7 Image-segmentation results for 12 head and neck muscles.** Image columns from **a** to **f** are the muscles in the left region of the head, and image columns from **g** to **l** are the muscles in the right region of the head. Image columns from left to right: sternocleidomastoid (SCM), trapezius (TZ), splenius capitis (SC), semispinalis capitis (SSC), levator scapulae (LS), and obliquus capitus inferior (OCI). Yellow curves denote the boundaries obtained by the level-set algorithm DRLSE, magenta curves represent the contours segmented by the crack-propagation method, and green curves represent the manual segmentation boundaries.

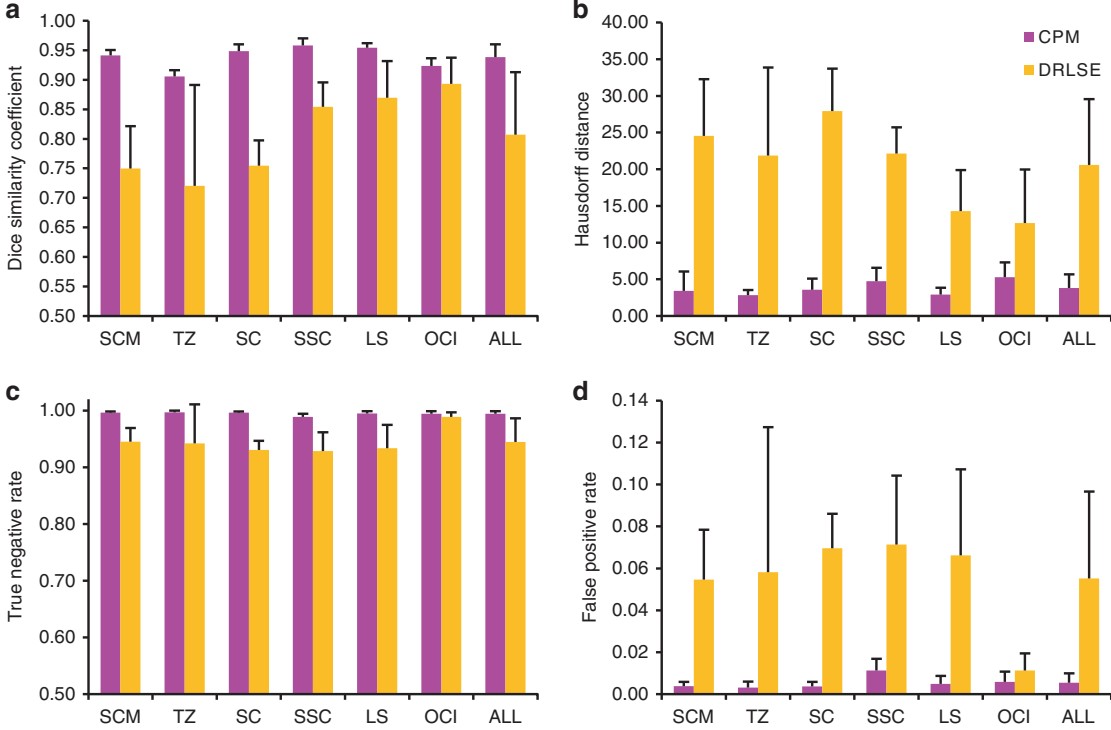

**Fig. 8 Comparison of the crack-propagation method (CPM) and the level- set algorithm DRLSE. a** Dice similarity coefficient, **b** Hausdorff distance, **c** true-negative rate, and **d** false positive rate. The statistical results are for muscles sternocleidomastoid (SCM), trapezius (TZ), splenius capitis (SC), semispinalis capitis (SSC), levator scapulae (LS), and obliquus capitus inferior (OCI). There are 10 images for each muscle group: five are from the left region and five are from the right region of the head. The last column combines all 60 images in six muscle groups. Magenta bars represent the CPM segmentation results and yellow bars represent the DRLSE. The error bars represent the standard deviation. Source data are provided in the Source Data file.

the level-set algorithm is affected by this region and cannot detect the correct boundary position. In contrast, CPM calculates correct boundaries for all images of the 12 head and neck muscles, clearly showcasing its great advantages in segmenting complex medical images with unclear and even discontinuous boundaries.

In order to quantitatively compare the performance of CPM and the level-set method, we compare several indicators of the segmentation results for six tissue groups in Fig. 8. Each of the groups includes 10 images from the same muscle, with five from the left region and five from the right region of the head (see Source Data). In addition to DSC and HD, we also compare the true-negative rate (TNR) and false-positive rate (FPR)[39]. TNR and FPR are used to describe the proportion of the correctly segmented non target region and the incorrectly segmented target region to quantify the boundary-overflow issue. The average values of the above four parameters are 0.9387, 3.7910, 0.9946, and 0.0054 for CPM, and 0.8078, 20.5753, 0.9448, and 0.0552 for DRLSE, respectively. The calculation results show that the mean DSC value is between 0.9058 and 0.9585, and the mean HD is between 2.8464 and 5.2710 for images of different muscle groups using CPM. However, the mean DSC value is between 0.7204 and 0.8934, and the mean HD is between 12.676 and 27.9274 when using DRLSE. It is clear that CPM is superior to DRLSE according to these performance metrics. A serious boundary overflow occurs in the segmentation process of DRLSE, whereas CPM does not have this problem. Currently, the accurate segmentation of the head and neck muscle tissues is a big challenge due to the lack of sufficient boundary information[40]. We have found very few publications that have attempted automatic segmentations for these muscles[11,12]. Although the methods used in these two studies are different from CPM, we can still evaluate CPM

performance by comparing the available indicators between CPM and the literature. Our average DSC and HD for segmentations of all 60 images of 12 head and neck muscles are 0.9387 and 1.6290 mm, respectively, if using mm as the unit of HD. The average DSC and HD in ref. [11] are 0.9191 and 6.4700 mm, respectively, and the average DSC in ref. [12] is 0.8500. Therefore, CPM has a higher accuracy than both of these two studies do.

**Automatic three-dimensional structure reconstruction.** One the most important applications of medical image segmentation is the three-dimensional reconstruction of tissue structures. As an example, we selected 12 consecutive MR images to automatically reconstruct the three-dimensional structure of the splenius capitis using CPM. One can see from Fig. 9 that the three-dimensional structure built using the two-dimensional boundaries extracted by CPM is highly consistent with the structure built based on the manual segmentation, which strongly highlights the advantages of CPM in the automatic segmentations of medical images involving complex features such as soft tissues with blurred boundaries.

## Discussion

In this study, we view image segmentation from a unique perspective, and propose the CPM, which applies the principles of fracture mechanics to image segmentation. The establishment of this method begins a bold new direction in the field of image segmentation. Using seemingly unrelated interdisciplinary knowledge, we make a breakthrough in solving a difficult problem in image processing and open a new way forward for the segmentation of complex medical images and the three-dimensional structure reconstruction.

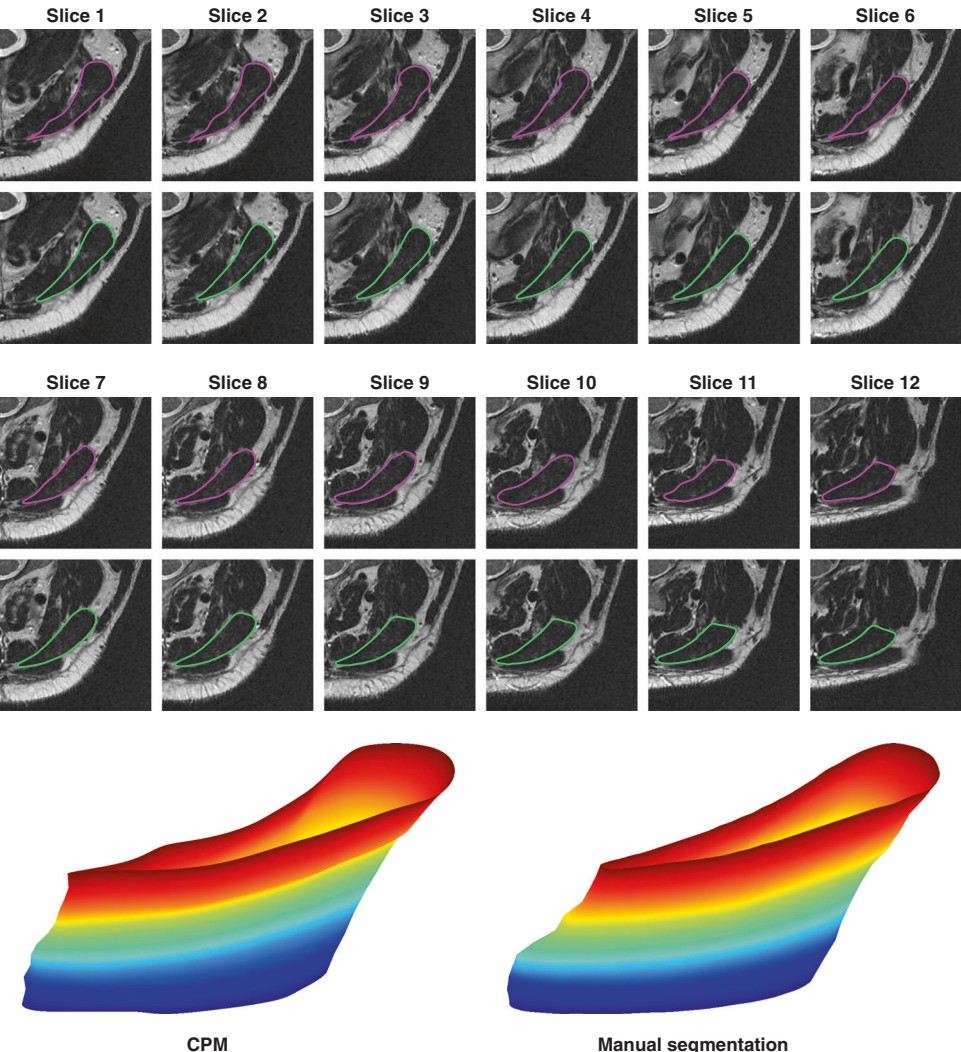

**Fig. 9 Segmentations of 12 consecutive magnetic resonance images of splenius capitis and reconstructions of the three-dimensional structure.**
Magenta curves in the images represent the final contours obtained by the crack-propagation method and green curves denote the manual segmentation boundaries. The red area of the three-dimensional structure corresponds to the first-layer image close to the head and the blue area corresponds to the last-layer image close to the neck.

In recent years, deep-learning algorithms have begun to out-perform previous state-of-the-art traditional methods and are gaining more popularity in research[15]. While applying to the segmentation of a wide range of images, a big challenge for deep-learning approaches is the scarcity of annotated training data, particularly for medical imaging applications, where both data and annotations are expensive to acquire[41]. Within systems lacking transparency, deep-learning networks require a great amount of hyperparameter tuning. Small changes in the hyper-parameters can result in disproportionately large changes in the network output[16]. Moreover, there is a known problem with deep neural networks, where visually indistinguishable images can return significantly different results. The interconnected complexity of the networks makes such issues difficult to trouble-shoot[42]. Therefore, current deep-learning methods have limitations when dealing with segmentations of some complex medical images. The CPM established in this study is a clear and convenient method that requires less input information. It cannot only segment common computer-synthesized images and medical images correctly, but also obtain the same results as manual segmentation when dealing with images involving the identifi-cation of fuzzy muscle boundaries. Especially for those blurred or even discontinuous edges that are difficult to deal with using existing automatic methods, CPM has unique advantages. There are currently large demands for complex image segmentation in clinical practice and research, for example, in the pathogenesis of sleep-disordered breathing and personalized surgical planning for obstructive sleep apnea patients[43–45], as well as the diagnosis and treatment of shoulder- and neck-pain diseases[46]. This unique segmentation method based on mechanical principles brings new insights into the study of image processing and pro-vides an important tool for solving difficult image-segmentation problems.

As a first step, we aim to propose and validate the feasibility of this unique idea. Therefore, we are not particularly concerned with optimizing the process of the crack propagation in this study. A commercial software for finite-element analysis is used directly for convenience in the crack calculation, and the crack-growth process will inevitably be limited by certain software configurations due to the software's requirements of adaptability to solve a wide range of problems, not just this one. In the next step, we will focus on an in-depth study on the algorithm of crack propagation, including simplifying the crack-calculation model based on the characteristics of the target boundary, and

optimizing the propagation process. We firmly believe that with the ever-increasing dependence on and demand for medical imaging in the diagnosis and treatment of diseases, the clear need for automatic recognition of complex tissues, and the importance of three-dimensional structure reconstruction to aid in the investigation of disease mechanisms and the development of effective treatments, CPM will continue to demonstrate its unique charm and vast potential.

## Methods

All equations and calculations involving matrices in the Methods section are element-wise.

**Local crack-propagation method (LCPM).** The first step of CPM is to convert the image into a thin-plate structure in order to induce crack propagation along grooves under highly concentrated stress. In fact, we do not need to transform the whole image into a mechanical model at once. Instead, we decompose the segmentation problem into a continuous evolution process in which the crack grows only in a local area. This process iteratively converts the partial image corresponding to a small neighborhood near the crack tip into a thin plate and calculates the crack expansion in the local region only. Following the direction of crack propagation of the current iteration of the local model (i.e., the direction of the boundary in the partial image), we determine the next iteration and local region. The new region partially overlaps with the previous one in that it contains a small portion of the crack, which had been generated in the previous local model, as the initial crack. Thus, cracks generated in the previous region will continue to expand in the new local model, as shown in Fig. 10. Finally, all the cracks generated in local models form a complete crack after the iterative process, resulting in the exact

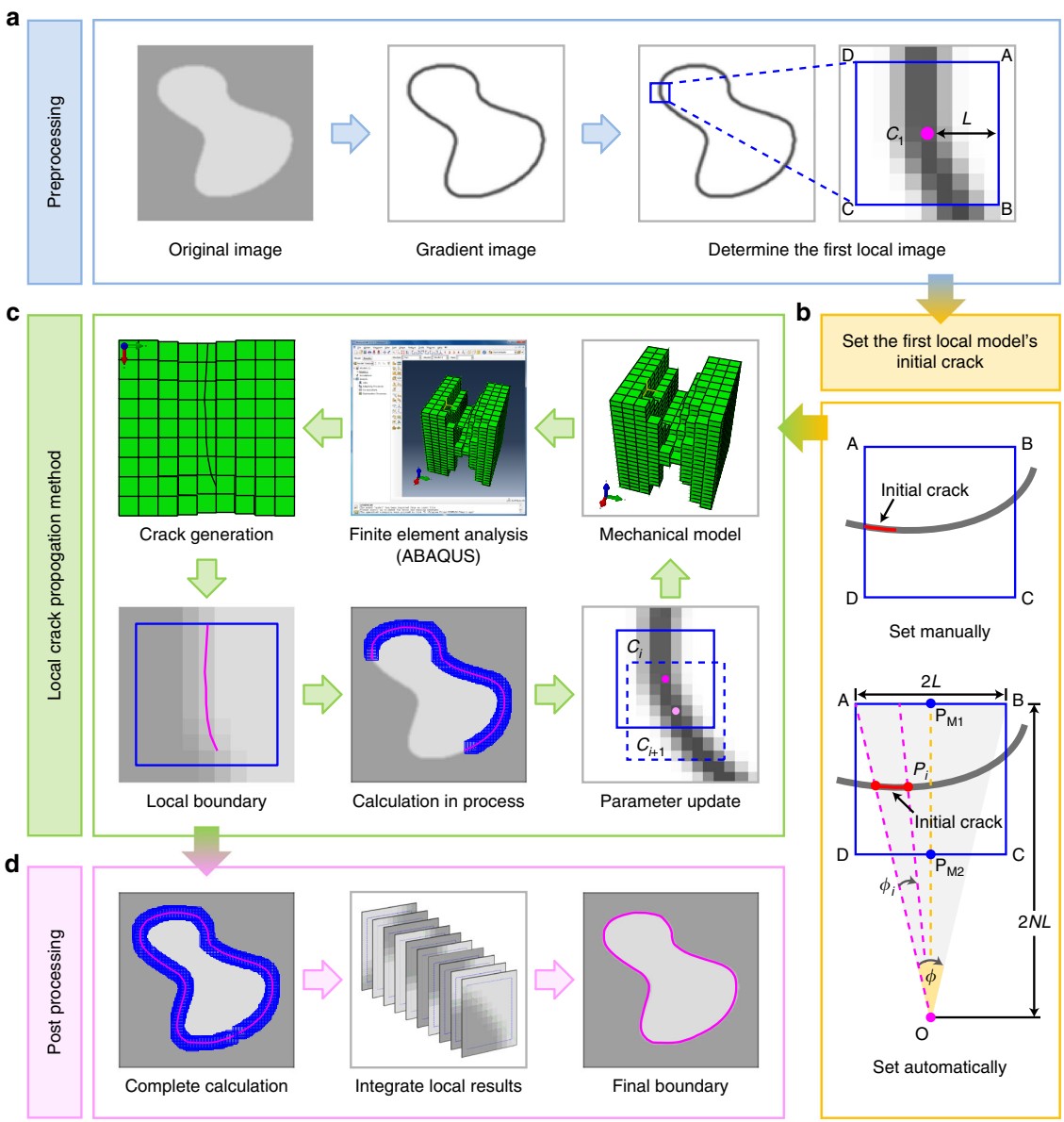

**Fig. 10 Pipeline of the local crack-propagation method algorithm. a** Preprocessing (blue box): Import the segmentation image and obtain the corresponding grayscale-gradient image. Determine the center $C_1$ of the initial partial image and the size of the local model $2L$. **b** Set the initial crack (yellow box) in the first local model manually (top half) or automatically (bottom half). **c** Local crack-propagation method (green box): establish the mechanical model based on the local image, set parameters, and boundary conditions of the finite-element analysis (FEA). Call FEA software ABAQUS to simulate crack propagation. Obtain the crack generated in the model. Calculate the local boundary in the partial image according to the crack. If the crack calculation of the entire image has not been completed, determine the next local model center $C_{i+1}$ based on the current center $C_i$ and the crack direction, and then establish the new local model to continue the crack propagation. **d** Post processing (pink box): After completing the crack calculation for the entire target area, integrate cracks generated in all local models to obtain the complete image boundary.

boundary position of the target area. We can call this method, which achieves image segmentation by splitting the crack propagation of the overall model into multiple local models, the local crack-propagation method (LCPM).

The LCPM has obvious advantages. The evolution process of local cracks can greatly reduce unnecessary and repetitive computations, and significantly improve image-segmentation efficiency. Converting the target area directly into a global model for crack-propagation simulation consumes a high amount of computation time. In reality, areas far away from current crack tip have no substantial effect on crack propagation, and including these areas in each calculation greatly increases computational cost for little gain. Although we need to simulate crack growth of multiple small regions separately, the number of displacement equations is drastically reduced due to the large reduction of nodes, greatly saving calculation time. Additionally, since the entire image contains a lot of grayscale information and the overall grayscale range of the pixels is large, it is often difficult to balance the grayscale of the entire original image to establish a complete mechanical model with obvious height differences between grooves and surrounding structures. The LCPM avoids this problem easily. The conversion from image to plate structure is based on the grayscale value in a small local area, so that any grayscale transformations to increase contrast can operate without interference from pixels in other regions, and we can easily amplify the grayscale differences near edges. This will maximize the effective height difference between grooves and adjacent structures in the mechanical model and strengthen stress concentration at grooves under external load. Finally, LCPM can also transform the dynamic load applied to the overall model into static loads applied to each local model, which simplifies the complexity of applying load in finite-element analysis (FEA).

The main steps of segmenting target regions by LCPM are as follows (Fig. 10). First, we import the segmentation image and obtain the corresponding grayscale-gradient image. We execute a nonlinear contrast-enhancing transformation on the gradient image to optimize image quality and highlight the edge information of target areas. We determine the position of the local calculation region in the original image and set relevant geometric parameters of the local model. Then, we establish the mechanical model based on the transformed local image, determine the position of the initial crack, and set parameters and boundary conditions of FEA. We execute the FEA software to simulate crack propagation. Once we obtain the edge position in current partial image, we update the relevant parameters according to the edge direction and propagate the next iteration of the model along the edge direction. We continue to simulate crack propagation until the crack expansion is complete at all positions in the entire target region. Finally, we integrate all the results of the partial models to obtain a complete outline of the target region.

### Intensity transformation

A mechanical model is established based on the grayscale distribution obtained by convolution of the original image and a Gaussian function combined with a gradient calculation. Let $I_o$ be the grayscale distribution of an original image. We define the new grayscale distribution $I_G$ after convolution and gradient operation by

$$I_G = |\nabla(G_\sigma * I_o)|^2 \tag{1}$$

where $G_\sigma$ is a Gaussian kernel function with standard deviation $\sigma$, * represents the convolution operation, and $\nabla$ is the gradient operator. According to the properties of convolution, $\nabla(G_\sigma * I_o) = (\nabla G_\sigma) * I_o$, we can reduce the computation time greatly by writing

$$I_G = |\nabla G_\sigma * I_o|^2 \tag{2}$$

The convolution in Eq. (2) is used to smooth the original image to reduce noise and optimize the quality of the image.

Since the gradient function reflects the change in the original grayscale distribution, $I_G$ usually takes larger values at the boundaries of the target region than the nonboundary area. In order to map the object edge to the groove of plate structure, we use a negative transformation to scale the values of $I_G$ to the interval [0, 1]

$$\widehat{I_G} = 1 - \frac{I_G - I_{Gmin}}{I_{Gmax} - I_{Gmin}} \tag{3}$$

where $I_{Gmin}$ and $I_{Gmax}$ represent the minimum and maximum values in $I_G$. After Eq. (3), the new grayscale distribution function $\widehat{I_G}$ takes on smaller values at object boundaries. When we convert $\widehat{I_G}$ into the thickness distribution of a thin plate, a groove structure forms at the boundary positions.

In order to increase the height difference between the grooves and surrounding structures, we perform a power-law transformation[47] on $\widehat{I_G}$. The grayscale distribution of transformed image $I_M$ is

$$I_M = \widehat{I_G}^{\gamma} \tag{4}$$

where $\gamma$ is constant. If $\gamma > 1$, then higher grayscale values are stretched and the output image appears darker than the input. If $\gamma < 1$, then the opposite happens, with lower values being stretched and the resulting image appearing brighter. By choosing different values of $\gamma$ to selectively stretch a certain grayscale interval, we can deepen the grooves corresponding to the object boundaries and improve the quality of the inhomogeneous image.

### Local model setup

We need to determine several geometric parameters when building the plate model from a partial image. These include the pixel coordinates of the local image center in the original image, the size of the partial image, and the maximum thickness of the model when transforming the partial image into a plate structure. Only the first local model center needs to be predetermined, as subsequent local area center positions can be automatically calculated from the cracks generated in the previous model.

Let $C(x_i, y_i)$ be the center of the $i$th local image with size $2L$. The grayscale distribution $I_{Mi}$ in this local region can be defined by

$$I_{Mi}(x, y) = \{I_M(x, y), \text{where } \max(|x - x_i|, |y - y_i|) \le L\} \tag{5}$$

$(x, y)$ is the pixel coordinate. If $H$ is the maximum thickness of the local plate model, the height at the position $(x, y)$ in the local model is

$$h(x, y) = \frac{\left[I_{Mi}(x, y) - I_{Mi_{min}}\right](H - 1)}{I_{Mi_{max}} - I_{Mi_{min}}} + 1 \tag{6}$$

where $I_{Mimin}$ and $I_{Mimax}$ represent the minimum and maximum grayscale values in this partial image.

### Finite element

The mesh of the finite-element models in this paper is a three-dimensional extrapolation of the two-dimensional pixel layout of the original image. If the length of a pixel is $L_p$ and the area is $L_p \times L_p$, a cube whose height is also $L_p$ is defined as the element. The grayscale value is simply equal to how thick the model is in pixels in the vertical direction of pixel face based on Eq. (6) when converting from image intensity to plate thickness. This mesh-construction method has obvious advantages, as it can generate high-quality units simply, evenly, and quickly, while avoiding the difficulty of generating qualified meshes on the uneven surface of the mechanical model.

### Material properties

We use a relatively large Young's modulus and a Poisson's ratio close to 0.5. The former can make the structure immune to deformations when the crack is generated, and the latter keeps the material incompressible under external force and keeps the volume of elements constant. The fracture criterion selected in this study is the maximum principal stress criterion, and the material parameters used in FEA are as follows: Young's modulus 210 GPa, Poisson's ratio 0.49, maximum principal stress 220 MPa, and fracture energy per unit area 42,200 J m$^{-2}$.

### Initial crack

In each partial model, we set an initial crack. The initial crack will grow along grooves under external load until reaching object boundaries. For the first local model, the initial crack needs to be specified. After completing crack propagation in the current region, the next local model can be determined according to the position and the orientation of the crack. The new area will partially overlap the old one so that it can contain the end portion of the crack in the previous area as the initial crack.

The initial crack in the first local model can be provided either manually or automatically (Fig. 10b). In the manual method (top half of Fig. 10b), one needs only to choose a short line a few pixels in length along the target boundary as the position of the initial crack in the mechanical model. This selection is simple and accurate because the target boundary is generally very clear in such a location and there is an obvious groove corresponding to the target boundary in the mechanical model. For the automatic method (bottom half of Fig. 10b), the initial crack can be generated by the following procedure: define the side length of square ABCD representing the local image to be $2L$. The target boundary intersects with the two parallel edges AD and BC. Draw a straight line starting from $P_{M1}$, the midpoint of the side AB, and passing through $P_{M2}$, the midpoint of the side DC, ending at the point O. The distance from $P_{M1}$ to O is $N$ times the length of AB ($N > 1$). Therefore, a triangle AOB is formed with angle at point O equal to $\varphi = 2 \arctan(1/(2N))$. Dividing $\varphi$ by an integer $m$ defines the angle increment, $\Delta\varphi = 2 \arctan(1/(2N))/m$. Starting at line OA, and sweeping toward line OB by incrementing the angle by $\Delta\varphi$ each iteration, the ray with angle $\varphi_i = i \Delta\varphi$ ($i = 0, 1, 2, \ldots$) will intersect the target boundary at the point $P_i$ in this local image area. Because the grayscale value of the pixels at the target boundary changes significantly, an edge-detection operator, such as the Canny edge detector, can easily identify the coordinates of the points $P_i$. After detecting several boundary points, the initial crack of the first local model can be automatically formed by connecting these points to form a small line segment.

### Boundary conditions

The external force applied to the plate structure is near the crack tip and normal to the direction of the initial crack. This way of loading can increase stress concentration at the grooves and avoid introducing interference cracks at other high-stress areas caused by the existence of low-height regions between external force location and the initial crack.

The edges of the interested area are often irregular curves, so the grooves formed in the mechanical model will also change constantly. Therefore, we must continuously adjust the direction of force as the crack grows, which will increase the complexity of loading. The LCPM can effectively avoid this problem. Since the image size for each partial model is small, object boundaries falling into this area can be approximated as a straight line. In a local model, the direction of external

force can be fixed, and the dynamic boundary condition can be transformed into a series of static conditions. Thus, LCPM greatly simplifies the numerical simulation process. We want to point out that it does not need to specify the quantity of the external force applied to the structure during a simulation of the crack propagation using LCPM. We only need to set a maximum load in the code to ensure that it is large enough to break all thin-plate structures transformed from images. Starting from zero, the external force applied to the model will increase linearly with each iteration. Once the crack is generated on the current local model, the algorithm will stop increasing the load, and automatically move to the next local model region.

**The extended finite-element method (XFEM)**. After determining input parameters and conditions, the FEA software can be invoked to simulate crack growth. The FEA software used in this paper is ABAQUS (Dassault Systèmes Simulia Corp., Providence, RI, USA), and the numerical calculation method is XFEM[48,49]. Conventional finite- element methods (CFEM) usually use polynomial interpolation functions as the shape function, which requires continuity of displacement and material. When dealing with a strong discontinuous problem involving damage and failure, CFEM requires setting the crack as the element boundary and the crack tip as the node. Considerable mesh refinement is also needed in the neighborhood of the crack tip. Therefore, the mesh must be updated continuously as the crack progresses, which leads to low calculation efficiency. In contrast, XFEM is an extension of CFEM based on the concept of partition of unity[50], which introduces the special enriched functions in conjunction with additional degrees of freedom to ensure the presence of discontinuities in modeling a growing crack. Due to the independence of the mesh and the crack surface, XFEM does not need to update the mesh when modeling crack propagation.

Using XFEM to simulate crack propagation, the fracture criterion, the damage-evolution law, and the damage-stability coefficient need to be specified. The fracture criterion is used to control the onset of damage of the material, and the maximum principal stress-failure criterion is selected in this study[51]. An additional crack is introduced or an existing crack is extended when the maximum principal stress in a model reaches the given threshold. Damage-evolution parameters control the development of the crack, and a mixed-mode, energy-based damage-evolution law based on a power-law criterion is selected for damage propagation. The damage-stability coefficient is used to improve convergence performance.

**The extraction of local edge**. The crack information calculated in a local model can be converted into boundaries in the corresponding image. The ABAQUS output file contains a $\varphi$ value for each unit node. The magnitude of $\varphi$ indicates the spatial distance of the node to the crack surface, and the sign of $\varphi$ indicates which side of the crack surface the node is on. Given $\varphi$, we can calculate the specific position of the crack surface in the current model. The crack surface obtained is a curved surface in space, but the object boundary is a two-dimensional curve. Mapping the three-dimensional crack surface to a two-dimensional plane, we can obtain the edges of the interested region in the partial image.

**Parameter update**. Establishing a new local model based on the crack direction, the center of the next calculation region needs to be updated as the crack propagates. The boundary acquired by the crack of the $i$th local model $M_i$ is

$$S_i = \left[ (x_1, y_1), (x_2, y_2), \dots , (x_j, y_j), \dots , (x_n, y_n) \right] \qquad (7)$$

where $(x_j, y_j)$ are the coordinates of points in $S_i$. Since the edge of the partial image corresponding to a local model can be approximated as a straight line, the slope $k_i$ of $S_i$ can be obtained by fitting the points in $S_i$. If the center of a local model $M_i$ is $C_i(x_i, y_i)$, the center $C_{i+1}(x_{i+1}, y_{i+1})$ of $M_i$ can be expressed as

$$C_{i+1}\left(x_{i+1}, y_{i+1}\right) = (x_i \pm \alpha \cos \theta_i, y_i \pm \alpha \sin \theta_i) \qquad (8)$$

where $\theta_i = \arctan(k_i)$ and $\alpha$ is the distance between two adjacent models. If we require that $0 < \alpha < 2L$, we can ensure that there is an overlapping area between adjacent local models, so that the new model can include a partial crack of the previous model as an initial crack to simulate crack propagation.

**Reporting summary**. Further information on research design is available in the Nature Research Reporting Summary linked to this article.

## Data availability
The authors declare that the data supporting the findings of this study and the source data underlying Figs. 6a, b and 8a–d are available in the attached Source Data file.

## Code availability
The custom codes that support the findings of this study are available from the corresponding author upon reasonable request.

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

## Acknowledgements

This work is supported by grants from the National Natural Science Foundation of China (31670959 and 81171422), and the National Science and Technology Pillar Program of China (2012BAI05B03).

## Author contributions

Y.H. conceived the project. Y.H. and G.H. designed the study. G.H. performed data collection and numerical simulation. Y.H., G.H., C.J., and H.X. analyzed and interpreted the data. Y.H. and G.H. wrote the paper with contributions from all authors.

## Competing interests

The authors declare no competing interests.
