## [Peer Review File · Nature Communications]

Reviewers' comments:

Reviewer #1 (Remarks to the Author):

This paper presents an automatic image segmentation based on the crack propagation method (CPM), based on the principles of fracture mechanics. This method converts the image segmentation problem into a mechanical problem. The boundary information of the target area are extracted by tracing the crack propagation on a thin plate with grooves corresponding to the area edge. Authors claim is that the CPM achieve good results by the segmenting images involving blurred or even discontinuous boundaries, a task difficult to achieve by existing auto-segmentation methods. However, I still have some concerns.

1. How the initialization of your model is done? Is it manual or automatic?
2. In the comparison pipeline did you use the same initialization for all the methods tested to get a fair comparison? Authors should show the initialization of the method to highlight the complexity of the segmentation task.
3. Another point not clear is the method tuning. How is it done?
4. How can you quantify the external force applied to the plate structure?
5. A graphical representation of steps (methods and software) will improve the paper comprehension

Reviewer #2 (Remarks to the Author):

This paper presents a new Crack Propagation Method (CPM) for medical image segmentation. This method is inspired by the way glass cracks and extracts boundary by tracking the crack propagation on a thin plate. The method is evaluated on simulation and real medical images. Better performance over level-set is demonstrated.

Overall the method is interesting and the paper is well presented. There are several issues that need to be addressed.

1. Result evaluation is quite limited. Quantitative evaluation is presented in Figure 6, which shows results on six images only. This extent of evaluation is very little. A lot more images should be used in quantitative evaluation, and these images should show considerable level of diversity.
2. Figure 5 shows visual results on different types of medical images. Quantitative results should be included as well, and the results should be based on a large set of images, not just one image per type of image.
3. Throughout the paper, the authors seem to use the term "image" to refer to a 2D image slice. This is especially clear when Figure 7 is presented. In fact, the term "image" is more commonly used to refer to a whole 3D image scan of a patient. So changes need to be made throughout the paper. And this could affect the results in Figure 6 as well, if evaluation was only done a single 2D slice per case, which would further show the insufficiency of evaluation.
4. The proposed method requires an initial crack. It's not clear how the initial crack looks like and how accurate it needs to be. This could have a real influence on the segmentation result. Visual illustration of the initial crack and quantitative evaluation of its effect on the segmentation result should be presented.

AUTHOR'S RESPONSE IN BLUE:

Reviewer #1 (Remarks to the Author):

This paper present an automatic image segmentation based on the crack propagation method (CPM), based on the principles of fracture mechanics. This method converts the image segmentation problem into a mechanical problem. The boundary information of the target area are extracted by tracing the crack propagation on a thin plate with grooves corresponding to the area edge. Authors claim is that the CPM achieve good results by the segmenting images involving blurred or even discontinuous boundaries, a task difficult to achieve by existing auto-segmentation methods. However, I still have some concerns.

1. How the initialization of your model is done? Is it manual or automatic?

RESPONSE:

A general initialization of the CPM, including the size of the local model, material properties, criteria of crack generation, and more, is achieved by setting fixed parameters in the code. A key initialization of the CPM is to set the initial crack in the first local model for a crack propagation along the entire target boundary. The initial crack in the first local model can be provided either manually or automatically. We have added a new paragraph in the section “initial crack” in Methods and a subfigure 10b to address the crack initialization in detail in this revised manuscript.

2. In the comparison pipeline did you use the same initialization for all the methods tested to get a faire comparison? Authors should show the initialization of the method to highlight the complexity of the segmentation task.

RESPONSE:

Yes, to get a fair comparison, we use the same images with the same preprocessing procedures, such as to reduce noise and transform the original image to a gradient image, before any particular segmentation methods are applied. However, there are also special initialization requirements for different segmentation methods. The level set algorithm is a method based on the area extension, and therefore its initialization needs to set a closed curve within or enclosing the target area. The CPM is a method to track the crack propagation along the target boundary directly, and therefore its initialization is to provide a small crack on the target boundary rather than a closed curve inside or outside the target area. The comparison among different methods is indeed fair even though particular initializations, determined by each particular segmentation method, may be somewhat different as mentioned above. We have provided many more details on the initialization of the CPM in this revised manuscript now.

3. Another point not clear is the method tuning. How is it done?

RESPONSE:

This method can go very smoothly after setting the input parameters described in the manuscript and the initial crack. So far we have not experienced any particular tuning requirements.

4. How can you quantify the external force applied to the plate structure?

RESPONSE:

We actually do not need to specify the quantity of the external force applied to the structure during a simulation of the crack propagation. All we need to do is to set a maximum load in the code to ensure that it is large enough to break all thin plate structures transformed from 2D images. Starting from zero, the external force applied to the model will increase linearly with each iteration. Once the crack is generated on the current local model, the algorithm will stop increasing the load, and automatically move to the next local model region. We have added some explanations about this into the revised manuscript to make it clear.

5. A graphical representation of steps (methods and software) will improve the paper comprehension

RESPONSE:

Thanks for the suggestion. Figure 10 in this revised manuscript, modified from Fig. 7 in the previous manuscript, is the graphical representation of all CPM steps, including the manual and automatic methods to set the initial crack in the first local model and the software used in the finite element analysis of the crack propagation. Hope this figure to be helpful for the reader to understand the procedures of the image segmentation using CPM.

Reviewer #2 (Remarks to the Author):

This paper presents a new Crack Propagation Method (CPM) for medical image segmentation. This method is inspired by the way glass cracks and extracts boundary by tracking the crack propagation on a thin plate. The method is evaluated on simulation and real medical images. Better performance over level-set is demonstrated.

Overall the method is interesting and the paper is well presented. There are several issues that need to be addressed.

1. Result evaluation is quite limited. Quantitative evaluation is presented in Figure 6, which shows results on six images only. This extent of evaluation is very little. A lot more images should be used in quantitative evaluation, and these images should show considerable level of diversity.

RESPONSE:

Thanks for the comments. We have added many more images for quantitative evaluation. Compared to Figure 6 in our previous manuscript, which shows the test results of 6 images from 6 muscles, the new Figures 7 and 8 in this revised manuscript include results of 60 images from 12 head and neck muscles with a considerable level of diversity.

2. Figure 5 shows visual results on different types of medical images. Quantitative results should be included as well, and the results should be based on a large set of images, not just one image per type of image.

RESPONSE:

Thanks for the suggestion. In the new Figure 5, we have increased the visual results from 5 medical images in the old Figure 5 in the previous manuscript to 11 types of medical images on different human organs and tissues obtained using different imaging techniques, such as MRI, CT, X-ray, and ultrasound methods. In the newly added Figure 6 in this revised manuscript, we show the quantitative results of 67 medical images regarding these 11 types of human organs and tissues.

3. Throughout the paper, the authors seem to use the term "image" to refer to a 2D image slice. This is especially clear when Figure 7 is presented. In fact, the term "image" is more commonly used to refer to a whole 3D image scan of a patient. So changes need to be made throughout the paper. And this could affect the results in Figure 6 as well, if evaluation was only done a single 2D slice per case, which would further show the insufficiency of evaluation.

RESPONSE:

Sorry for the confusion. We do use the term "image" for a 2D image slice in this paper, which is convenient for us to describe the image segmentations using CPM.

We have now added a sentence ‘The term “image” used throughout the paper denotes a two-dimensional image slice if not specified.’” at the beginning of the Results section in this revised manuscript to avoid confusion. The current evaluation shown in the new Figures 7 and 8 involves 11 types of human organs and tissues across a total of 67 two-dimensional images now.

4. The proposed method requires an initial crack. It's not clear how the initial crack looks like and how accurate it needs to be. This could have a real influence on the segmentation result. Visual illustration of the initial crack and quantitative evaluation of its effect on the segmentation result should be presented.

RESPONSE:

The first local model in CPM requires an initial crack, which can be provided either manually or automatically. Both ways can set the initial crack very accurately and easily. We have added a new paragraph in the section “initial crack” in Methods and a subfigure 10b to address the crack initialization in detail in this revised manuscript.

REVIEWERS' COMMENTS:

Reviewer #1 (Remarks to the Author):

The paper sounds novel and interesting for the the community. As all of my concerns have been resolved, I recommend accepting this paper.

Reviewer #2 (Remarks to the Author):

The revision has addressed the reviewer's comments.

AUTHOR'S RESPONSE IN BLUE:

Reviewer #1 (Remarks to the Author):

The paper sounds novel and interesting for the community. As all of my concerns have been resolved, I recommend accepting this paper.

RESPONSE:

Many thanks to the reviewer.

Reviewer #2 (Remarks to the Author):

The revision has addressed the reviewer's comments.

RESPONSE:

Many thanks to the reviewer.